

# Comments on "Cavitons and spontaneous hot flow anomalies in a hybrid-Vlasov global magnetospheric simulation" by Blanco-Cano et al. (2018)

Gábor Facskó[1]

[1]Rhea System GmbH., TIZ Building, Robert Bosch Str. 7., 64293 Darmstadt, Germany

**Correspondence:** Gábor Facskó (g.facsko@rheagroup.com)

Blanco-Cano et al. (2018) analysed the output of the VLASIATOR global hybrid Vlasov solver and intended to find Spontaneous Hot Flow Anomalies (SHFA, Zhang et al., 2013; Omidi et al., 2013). This is a very nice paper about the development of the foreshock cavitons and magnetosheath cavities based on unique global hybrid-Vlasov simulations. However, the simulation results cannot reproduce the main features of the SHFAs. The SHFAs (and the HFAs) show density and magnetic field magnitude drops in their cavity. The magnetic field is turbulent in the cavity and shocks form on both sides of the event. The temperature is very high in them, a few million K. The solar wind direction turns away from the radial direction and slows down (Facskó et al., 2010; Zhang et al., 2013; Omidi et al., 2013). The latter features gave the name of the phenomenon hence I had serious concerns about whether the Authors had detected SHFA in the paper above.

On Figure 3, 7, 9 of the paper above, the Authors see density and magnetic field drops. I can see no drop in either the density or magnetic field magnitude. Hence this event cannot be an SHFA or even a foreshock cavity (Sibeck et al., 2002). The simulated phenomena is not significantly hotter than the surrounding foreshock plasma. The foreshock plasma temperature is never observed at 10 million K. Hence locating in the foreshock cannot be an excuse for the missing feature of the phenomena. The Authors also see "*[. . . ] deviations from the bulk solar wind velocity are observed throughout the foreshock, and they are not prominent enough inside SHFAs to be unambiguously identified.*" The phenomenon that does not show anomalous flow cannot be called Spontaneous Hot Flow Anomaly.

Unfortunately, not only the physical indicators are missing from the simulation result, but the footprints and the signs of the SHFA formation processes too. The (S)HFAs are formed by the interaction of the solar wind ions with the reflected and accelerated ion beam of the bow shock. In young (S)HFAs, the two populations can be distinguished by the ion velocity distributions at $V_x = 0\,km/s$ and $V_x = 600\,km/s$ (see Figure 4b in Lucek et al. (2004) and Figure 7b in Zhang et al. (2010)). The c, h and n events on Figure 6 must be young (at least c; as the mature (S)HFAs have no such velocity distributions). I cannot see the typical distribution with double peak. Hence, these structures are not SHFAs.



The SHFAs are surrounded by shocks. Their presence proves that the cavity is not in equilibrium and expands. If the VLASIATOR cannot create them, that is big a problem. The hybrid simulations of Omidi et al. (2013) could present these shoulders (Lin (2002) could also have simulated them for HFAs). Furthermore, these shocks lead to the observed depletion of the solar wind velocity because the deceleration of the solar wind comes from the bad fitting and plasma moment calculation (Parks et al., 2013; Kecskeméty et al., 2006, Figure 3, 7). Hence, it is possible to explain the missing solar wind deceleration

5    if these shocks are present. If both features are missing, the phenomena cannot be SHFA or the VLASIATOR needs further improvement to be able to study them.

The Authors also study foreshock cavitons, magnetosheath filaments and structures in this paper above. My comments are limited only for the identification and analysis of the "SHFA events" of the simulation. Based on the remarks described above, I am sure that the features in the VLASIATOR simulations are not SHFAs.

*Acknowledgements.* Gábor Facskó thanks Sophie Burley for improving the English of the paper.





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
