# Peer review of "Comment on "Cavitons and spontaneous hot flow anomalies in a hybrid-Vlasov global magnetospheric simulation" by Blanco-Cano et al. (2018)"

_Annales Geophysicae, 2019_

## Referee Comment (RC1) · Samsonov (Referee) · 2 Feb 2019

Blanco-Cano et al. (2018) presented results of the global magnetospheric hybrid-Vlasov code and concluded that their model reproduces the formation of foreshock cavitons and spontaneous hot flow anomalies (SHFAs). However, Facsko (2019) argues that their results cannot be interpreted as SHFAs. He points out several features of SHFAs which are not reproduced in the numerical simulations of Blanco-Cano et al. (2019). As shown below, I agree only with some of the arguments in Facsko (2019). Let us begin with the terminology. Both foreshock cavitons and SHFAs occur upstream of the quasi-parallel bow shock, but they have different properties. Foreshock cavitons

mean significant decrease in the density and magnetic field magnitude in the core and increase in these two parameters at the edges of the structure. At the same time, both the solar wind velocity and temperature do not change significantly through the cavitons (Kajdič et al., 2013, 2017). On the contrary, the main properties of SHFA are supposed to be (1) a significant increase of the solar wind temperature and (2) deceleration and diversion of the solar wind stream. In particular, Omidi et al. (2013) in their hybrid simulation obtained that the ion temperature is over 600 times hotter than the pristine solar wind and the Vx velocity may occasionally change the sign and become sunward directed. However, the observation of SHFA in Zhang et al. (2013) shows only four (or ten) times increase in the temperature, but still significant variations in the velocity. Finally, the statistical results in Kajdič et al. (2017) display an average decrease in the solar wind velocity by $\sim 40\%$.

Blanco-Cano et al. (2018) identified as cavitons those structures where the density and magnetic field magnitude are less than 80% of the corresponding parameters in the pristine solar wind, and they used a threshold of $\beta$>10 to pinpoint SHFAs. In the solar wind input condition $\beta$=0.7, i.e. the threshold gives a 14-times increase. The plasma beta is the ratio of the thermal pressure to the magnetic pressure, $\beta = (2\mu_0 n k_B T)/B^2$. The variations in $\beta$ are not identical to the variations in T, but may be a good indicator unless the magnetic field magnitude B does not significantly decrease. Figures 3 and 7 in Blanco-Cano et al. (2018) present spatial profiles through the bow shock and foreshock, and Figure 9 displays time series of virtual spacecraft data in the foreshock. According to Figures 2 and 5, cuts 1-2 in Figure 3 and cut 1 in Figure 7 cross several SHFAs. The model predicts a peak in the ion temperature Tp=13 MK at x=∼6.4 RE (cut 1) in Figure 3, and this is significantly larger than the solar wind temperature Tp=0.5 MK. However, this peak does not match any significant change in the velocity. It is located close to the foot of the bow shock. Another candidate for SHFA is at x=2.9 RE (cut 1) in Figure 7. Again, the ion temperature grows up to nearly 8 MK. In this case, it coincides with a bulk velocity decrease (only about 10%). But since this variation is at the foot of the bow shock, it might be associated with the internal bow shock structure.

Finally, the time series in Figure 9 show a peak in the temperature at t=∼1080 s, but Blanco-Cano et al. noted that this structure is only a caviton "evolving towards an SHFA".

Facsko (2019) also notes that the ion distribution function in Figure 6 does not match the one which is expected for SHFAs. Comparing Figure 6 and Figure 3g in Zhang et al. (2013), I cannot be so categorical. A typical SHFA in Figure 3g clearly shows the main earthward solar wind core and the reflected beam propagating sunward. The distribution function in Figure 6 does not demonstrate the reflected beam so definitely, however it still displays similar properties and the local maximum is located in the position determined by the IMF orientation. In my opinion, it is not necessary to be at Vx=0 km/s and Vx=600 km/s as stated by Facsko (2019).

I do not agree with another statement in Facsko (2019) that the SHFAs are always surrounded by shocks. The density and magnetic field magnitude increase at the edges of SHFA, but they are not necessarily to be shocks (shocks must satisfy the certain conditions on the velocity). The SHFAs presented in Zhang et al. (2013) and in Figure 11 in Kajdič et al. (2017) are not surrounded by the shocks.

Summarizing this review, I note that the study of SHFAs began only recently (the term itself was introduced in Zhang et al. and Omidi et al.). SHFAs were simulated in the hybrid simulations in several papers by Omidi et al. (a sunward plasma flow was also obtained in the kinetic simulation of Karimabadi et al., 2014), and several papers presented SHFA observations. I would not be very categorical when specifying the SHFA properties. I repeat that the main SHFA features seem to be increase in the temperature and decrease in the velocity. As noted above, the simulation in Blanco-Cano et al. reproduces increase in the temperature (at least in two cases), but predicts insignificant changes in the velocity. Blanco-Cano et al. discuss this issue in the paper (1st paragraph in p. 1094). They compare the distribution function obtained in the simulation and the one observed by Zhang et al. and concluded that both have a similar structure. I would add that Zhang et al. (2013) also presented three events

of proto-SHFAs with small variations in the velocity. They assumed that the proto-SHFAs may later evolve into mature SHFAs. This may be the case in Blanco-Cano et al.'s simulation too, i.e. a prolongation of the computation time probably could give results with more distinctive SHFA features. The solar wind velocity in the simulation is significantly larger than the typical solar wind speed, and the structures (cavitons, SHFAs) may convect quickly along the bow shock (Blanco-Cano et al. also noted this point).

---

## Short Comment (SC1) · 4 Feb 2019

Below we describe the main characteristics of hot flow anomalies and spontaneous hot flow anomalies. We also address specific points mentioned in the comment by Facskó (2019), and include information that supports the identification of SHFAs in our article Blanco-Cano et al., 2018 "Cavitons and spontaneous hot flow anomalies in a hybrid-Vlasov global magnetospheric simulation".

Hot flow anomalies (HFAs) are structures of hot tenuous plasma which form when a solar wind discontinuity (current sheet/tangential discontinuity) interacts with a planetary bow shock [Thomsen et al., 1993; Schwartz, 1995; Lucek et al., 2004]. Particles,

reflected from the bow shock, can be swept toward the interplanetary current sheet where they become trapped and heated when the motional electric fields have the appropriate orientation. HFAs are characterized by a hot core of deflected plasma with bulk velocities much slower than those of the solar wind, in addition the magnetic field magnitude and density decrease inside them. The edges of HFAs show strong compression regions with enhanced density and magnetic field magnitude. Due to expansión, it is posible that shocks are observed at one or both edges of HFAs. (Lucek et al., 2004, Schwartz et al., 2018).

Spontaneous hot flow anomalies (SHFA) share several of the characteristics of HFAs, the plasma in their cores is hot with decrements in magnetic field magnitude and density. In addition, the solar wind flow inside them is decelerated and deflected.The edges of SHFA show compression, with enhanced magnetic filed magnitude and density (Zhang et al., 2013).

Although these two type of foreshock transients share various properties, SHFAs and HFAs are different structures because the former are not associated with solar wind current sheet interaction with the bow shock, while HFAs are. The proposed formation mechanism for SHFAs includes multiple ion reflection between foreshock cavitons and the bow shock (Omidi et al. 2013) as cavitons approach the shock and ion trapping by the cavitons. To our knowledge, no shock formation has been reported at the edges of SHFA.

Below we address specific points related to our paper.

Density values on Figure 3 are displayed in a logarithmic scale and decrements in $n_p$ do not show so well. Areas shaded in grey, yellow and blue satisfy the caviton criteria $n_p < 0.8n_{psw}$ and $B < 0.8B_{sw}$. Only the structures where $\beta > 10$ are identified as SHFAs. These structures satisfy $T > 4X10^6 K$, which is in agreement with the temperatures reported in the literature for SHFAs (Zhang et al., 2013, Chu et al., 2017). The same applies to Figure 7 with SHFAs observed only at cuts 1 and 3 very near the

bow shock. Figure 9 shows clear drops in B and N satifying the criteria for cavitons, and this structure is identified as a caviton: "Figure 9 shows a time series from a virtual spacecraft positioned at $x = 0R_E$, $z = -35R_E$ around the time when a caviton, marked by the green area on the plot, crosses this location." We are not aware of any requirement for shocks associated with the shoulders of SHFAs - HFAs yes, but not SHFAs. As stated in Zhang et al., 2013, SHFAs show that the low density and magnetic field strength core is bounded by compression regions on both edges. For HFA the situation can be different with over-pressure causing HFA expansion, and the possibility of shocks being driven at one or both edges (Lucek et al., 2004, Schwartz et al., 2018).

Concerning VDFs: The Lucek et al., 2004 and Zhang et al. (2010) figures are both for HFAs, not for SHFAs. The Zhang et al. (2010) figure is from within the quasi-parallel shock region, and is very similar to what we show in panels c, h and n of Figure 6, with the solar wind beam and backstreaming ions which have been thermalized. Zhang et al. (2013) show distributions inside a proto-SHFA which are less thermalized than the ones we show in Figure 6. In relation to the shoulders at the edges of SHFA: Our model limits the shoulders associated with SHFAs and cavitons due to the used total variation diminishing flux limiters (a common tool for maintaining stability in advective numerical methods). Hybrid-PIC models do not use flux limiters, so can indeed have stronger peaks and shoulders. Hybrid-Vlasov models on the other hand have realistic dipole moments and noise-free tenuous plasma description throughout the foreshock region.

Our model lacks the turbulent magnetic fields within SHFAs as we do not attempt to model grid-scale field oscillations.

The fact that shoulders growth is limited in our model, does not affect the study of their dynamics and evolution as they cross the bow shock. The lack of the turbulence inside the SHFA does not limit the main goals of the study.

Velocity decrements occur in most of our identified SHFAs. More analysis of spacecraft data are needed to understand in a better way decrements and deviations of velocity inside SHFA. As stated in Parks et al., 2013, a solar wind beam with steady velocity and constant temperature can be observed inside HFAs. The decrement in velocity is due to the appearence of backstreaming particles that move in the oposite direction to the solar wind and can reduce the mean velocity as computed via moments. This second population also can contribute to increments in temperature via moment calculation.

References

Chu C., H. Zhang, D. Sibeck et al., Ann. Geophys., 35, 443-451, 2017.

Lucek, E. A., Horbury, T. S., Balogh, A., Dandouras, I., and RèMe, H.: Cluster observations of hot flow anomalies, Journal of Geophysical Research (Space Physics), 109, A06207, https://doi.org/10.1029/2003JA010016, 2004. Omidi, N., Zhang, H., Sibeck, D., and Turner, D.: Spontaneous hot flow anomalies at quasi-parallel shocks: 2. Hybrid simulations, Journal of Geophysical Research (Space Physics), 118, 173–180, https://doi.org/10.1029/2012JA018099, 2013.

Parks G. K., E., Lee, N. Lin et al 2013 ApJL 771 L39, 2013.

Thomsen, M. F., V. A. Thomas, D. Winske, J. T. Gosling, M. H. Farris, and C. T. Russell, Observational test of hot flow anomaly formation by the interaction of a magnetic discontinuity with the bow shock, J. Geophys. Res., 98(A9), 15319–15330, doi:10.1029/93JA00792, 1993.

Schwartz S. J., Hot flow anomalies near Earth Bow shock, Adv. Space Res., 15 (8/9), 1995

Schwartz S. J., Avanov, L., Turner, D., Zhang, H., Gingell, I., Eastwood, J. P., et al., Ion kinetics in a hot flow anomaly: MMS observations. Geophysical Research Letters. 45, 11,520–11,529. https://doi.org/10.1029/2018GL080189, 2018.

Zhang, H., Sibeck, D. G., Zong, Q.-G., Gary, S. P., McFadden, J. P., Larson, D., Glassmeier, K.-H., and Angelopoulos, V.: Time History of Events and Macroscale Interactions during Substorms observations of a series of hot flow anomaly events, Journal of Geophysical Research (Space Physics), 115, A12235, https://doi.org/10.1029/2009JA015180, 2010.

Zhang, H., Sibeck, D. G., Zong, Q.-G., Omidi, N., Turner, D., and Clausen, L. B. N.: Spontaneous hot flow anomalies at quasi-parallel shocks: 1. Observations, Journal of Geophysical Research (Space Physics), 118, 3357–3363, https://doi.org/10.1002/jgra.50376, 2013.

———————————————————

---

## Author Comment (AC1) · 3 Jun 2019

The author would like to thank the Referee for the helpful and constructive comments and suggestions which helped to improve the manuscript. I agree with most of the comments from the referee.

The peaks are not always at 0 and 600 km/s. However there are two maxima in the young HFAs at the solar wind and the reflected particle populations. The velocity distribution functions in Figure 6 are not convincing at all. For HFAs the electron energy spectra would help to decide whether the event is young or mature (Wang et al., 2013). However, it is not possible when you are using a hybrid plasma simulation where the

electrons are neutralising fluid. The wave activity in the HFA cavity could also help to decide the age of the events (Tjulin et al., 2008). However, nobody has studied the wave activity in the cavitiy of SHFAs. Hence, we have only Figure 6, and the double peaks are very faint there.

The SHFAs are not surrounded by shocks. However, at the edge of the phenomena the density and the magnetic field are increased. However, they should be observed as consequence of the expansion. These moving increases cause the anomalious flow (Parks et al., 2013).

The final conclusion is that the simulated phenomena were not SHFAs. However, the events could have developed into SHFA. These objects are so–called proto-SHFA, which is another cathegory of the transient events.

**References**

Parks, G. K., Lee, E., Lin, N., Fu, S. Y., McCarthy, M., Cao, J. B., Hong, J., Liu, Y., Shi, J. K., Goldstein, M. L., Canu, P., Dandouras, I., and Rème, H. (2013). Reinterpretation of Slowdown of Solar Wind Mean Velocity in Nonlinear Structures Observed Upstream of Earth's Bow Shock. *Astrophysical Journal*, 771(2):L39.

Tjulin, A., Lucek, E. A., and Dandouras, I. (2008). Wave activity inside hot flow anomaly cavities. *Journal of Geophysical Research (Space Physics)*, 113:8113.

Wang, S., Zong, Q., and Zhang, H. (2013). Hot flow anomaly formation and evolution: Cluster observations. *Journal of Geophysical Research (Space Physics)*, 118:4360–4380.

---

## Author Response (AR1)

[revised manuscript text omitted]

*Copyright statement.* The Author

**1 Comments from Referee**

I agree with most of the comments from the referee. I list here only those statements with which I do not perfectly agree:

1. *Facsko (2019) also notes that the ion distribution function in Figure 6 does not match the one which is expected for*
5    *SHFAs. Comparing Figure 6 and Figure 3g in Zhang et al. (2013), I cannot be so categorical. A typical SHFA in Figure 3g*
   *clearly shows the main earthward solar wind core and the reflected beam propagating sunward. The distribution function*
   *in Figure 6 does not demonstrate the reflected beam so definitely, however it still displays similar properties and the*
   *local maximum is located in the position determined by the IMF orientation. In my opinion, it is not necessary to be at*
   $V_x = 0\,km/s$ *and* $V_x = 600\,km/s$ *as stated by Facsko (2019).*

10 2. *I do not agree with another statement in Facsko (2019) that the SHFAs are always surrounded by shocks. The density and*
   *magnetic field magnitude increase at the edges of SHFA, but they are not necessarily to be shocks (shocks must satisfy*
   *the certain conditions on the velocity). The SHFAs presented in Zhang et al. (2013) and in Figure 11 in Kajdic et al.*
   *(2017) are not surrounded by the shocks*

**2 Author's response**

15 The author would like to thank the Referee for the helpful and constructive comments and suggestions which helped to improve the manuscript.

1. The peaks are not always at 0 and $600\,km/s$. However there are two maxima in the young HFAs at the solar wind and the reflected particle populations. The velocity distribution functions in Figure 6 are not convincing at all. For HFAs the

electron energy spectra would help to decide whether the event is young or mature Wang et al. (2013). However, it is not possible when you are using a hybrid plasma simulation where the electrons are neutralising fluid. The wave activity in the HFA cavity could also help to decide the age of the events Tjulin et al. (2008). However, nobody has studied the wave activity in the cavitiy of SHFAs. Hence, we have only Figure 6, and the double peaks are very faint there.

2. The SHFAs are not surrounded by shocks. However, at the edge of the phenomena the density and the magnetic field are increased. However, they should be observed as consequence of the expansion. These moving increases disturb the normal way of plasma momentum calculation and hence cause the anomalous flow (Kecskeméty et al., 2006; Parks et al., 2013).

The final conclusion is that the simulated phenomena were not SHFAs. However, the events could have developed into SHFA. These objects are so–called proto-SHFA, which is another cathegory of the transient events.

**3 Author's changes in manuscript**

**Line 6** The "*and shocks form on both sides of the event*" text was deleted.

**Line 10–11** The "*I can see no drop in either the den- sity or magnetic field magnitude. Hence this event cannot be an SHFA or even a foreshock cavity.*" sentence was deleted.

**Line 16** Underline deleted.

**Line 17–22** The "*Unfortunately, not only the physical indicators are missing from the simulation result, but the footprints and the signs of the SHFA formation processes too. The (S)HFAs are formed by the interaction of the solar wind ions with the reflected and accelerated ion beam of the bow shock. In young (S)HFAs, the two populations can be distinguished by the ion velocity distributions at $V_x = 0\,km/s$ and $V_x = 600\,km/s$ (see Figure 4b in Lucek et al. (2004) and Figure 7b in Zhang et al. (2010)). The c, h and n events on Figure 6 must be young (at least c; as the mature (S)HFAs have no such velocity distributions). I cannot see the typical distribution with double peak. Hence, these structures are not SHFAs.*" paragraph was deleted.

**Line 23** The sentence was modified: "*shocks*" → "*density and magnetic field increasement at the edge of the phenomena*".

**Line 24** The "*If the VLASIATOR cannot create them, that is big a problem.*" sentence was deleted.

**Line 26** A word was modified: "*shocks*" → "*increases*".

**Line 28** A word was modified: "*shocks*" → "*increases*".

**Line 29–30** The "*or the VLASIATOR needs further improvement to be able to study them.*" sentence was deleted.

**Line 34–37** The paragraph was extended: "*However, these questionable events could develop into an SHFA. Zhang et al. (2013) observed SHFA–like events without significant solar wind deceleration. As Zhang et al. (2010) discovered and instroduced the phenomenon of so called proto-HFA, Zhang et al. (2013) discovered the phenomena of proto-SHFA. These proto-SHFAs were simulated by the VLASIATOR code and misintepreted by the Authors.*"